# Reasoning Models Outperform Standard Language Models in De Novo Protein Design

**ChatGPT**
OpenAI

**Alfred Greisen**
Eastlake High School
`greisenalfred@gmail.com`

**Longfei Cong**

**Per Jr. Greisen**

**Claude**
Anthropic

**Sergey Ovchinnikov**
Massachusetts Institute of Technology
`so3@mit.edu`

## Abstract

We compared reasoning-enhanced versus standard large language models for de novo protein design using four-helix bundles as a benchmark. Testing five Chat-GPT variants with identical prompts, we discovered a dramatic capability divide: reasoning models (o3, o4-mini) achieved 44% and 20% success rates respectively, while all standard language models achieved 0% success using pLDDT > 75 as the threshold. Two of the top-scoring sequences were experimentally tested. One was validated via circular dichroism (CD) spectroscopy, confirming $\alpha$-helical structure. These results suggest reasoning capabilities, not just model scale, are critical for complex scientific tasks like protein design.

## 1 Introduction

The emergence of reasoning-enhanced large language models raises fundamental questions about AI capabilities in science. While standard language models excel at pattern recognition, reasoning models are designed for systematic problem-solving. Whether this architectural difference translates to meaningful performance gaps in scientific applications remains largely unexplored.

Protein design provides an ideal test case because it requires applying established design principles rather than memorizing patterns [1]. Four-helix bundles serve as excellent benchmarks due to their structural simplicity and well-understood design rules [2].

Here we present the first systematic comparison of reasoning versus standard language models for de novo protein design, revealing a striking capability divide with implications for AI-driven scientific discovery.

**Overview and novelty.**   Unexpectedly, a viable design emerged from the initial prompt without any iterative refinement. This observation suggests that contemporary large language models encode useful biochemical priors for helical bundle design, enabling direct sequence proposal from natural-language specifications. We report computational metrics (AlphaFold pLDDT) alongside experimental confirmation of stability and secondary structure by CD.

## 2 Methods

### 2.1 Computational Pipeline

We implemented a minimal AI-driven design workflow in which a human provided a natural-language specification for the target topology (e.g., a four-helix bundle). The AI system (ChatGPT [3, 4, 5])

generated candidate amino-acid sequences directly from the prompt without hand-crafted search or optimization. Candidate sequences were then evaluated with AlphaFold [6], and model confidence was summarized using pLDDT [7]. Designs passing a simple topology screen (visual inspection of helix packing and bundle formation) were advanced to experimental synthesis and validation by circular dichroism (CD) spectroscopy [8].

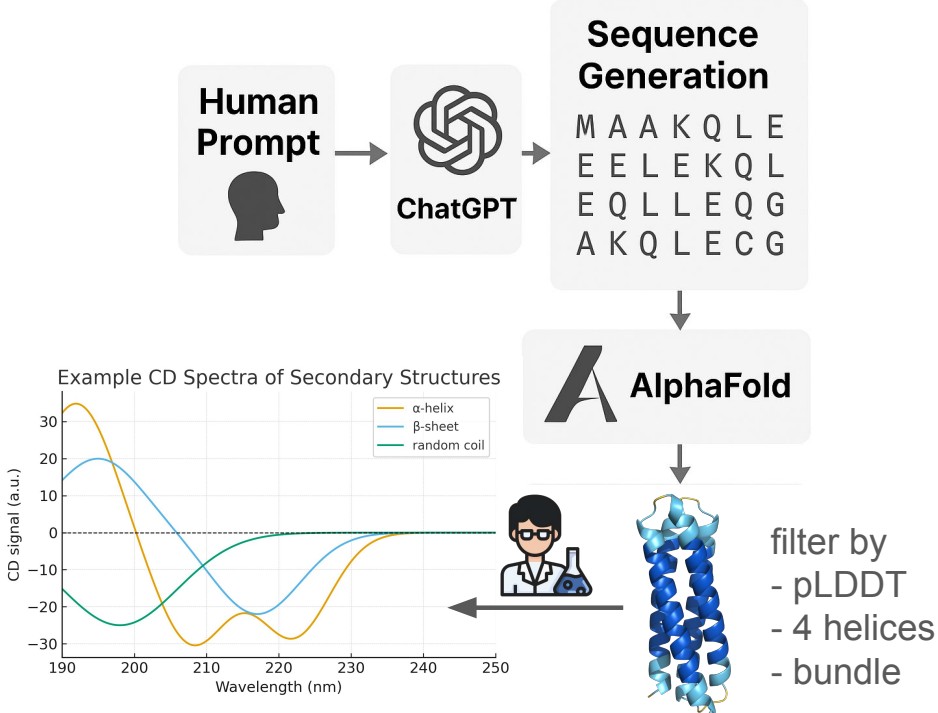

Figure 1: **Pipeline overview.** (i) Human provides the initial prompt; (ii) ChatGPT generates candidate sequences; (iii) AlphaFold evaluates structures and pLDDT; (iv) Human assesses helix bundle topology; (v) Experimental synthesis and CD validation.

We tested five ChatGPT variants: GPT-4o [3], o4-mini-high [4], o3 [4], GPT-4.5 [5], and o4-mini [4] (accessed through OpenAI API between November 2024 and January 2025).

Table 1: Overview of language models tested for protein design

| Model Type | Model | Architecture | N Sequences | Success Rate |
|---|---|---|---|---|
| Standard LLMs | GPT-4o | Transformer | 5 | 0% |
| | GPT-4.5 | Transformer | 4 | 0% |
| Reasoning Models | o3 | Chain-of-thought | 16 | 44% |
| | o4-mini | Chain-of-thought | 10 | 20% |
| | o4-mini-high | Chain-of-thought (high effort) | 5 | 0% |

Standard LLMs generate responses directly. Reasoning models use chain-of-thought processing with varying effort levels. Success rate refers to confident 4-helix bundles (pLDDT > 0.75).

Standard language models process prompts through direct pattern matching and generate responses in a single pass. In contrast, reasoning models utilize chain-of-thought processing to decompose the design problem into steps, enabling systematic application of biochemical principles. This architectural distinction is hypothesized to explain the performance gap observed in our results.

All models received identical prompts: *"Hey ChatGPT, can you give me a sequence of amino acids to code for a four-helical bundle using de novo design principles that can be pasted into AlphaFold?"*

Bundle formation was assessed by visual inspection of AlphaFold2 structures, requiring four helices in close proximity forming a compact arrangement.

Sequences were evaluated using AlphaFold2 [6] via the ColabFold implementation [9] with the following parameters: 0 recycles, single sequence input (no MSA), and model_2_ptm. We applied two success criteria:

1. **4-Helix Bundle**: Formation of four helices in bundle configuration

2. **Confident 4-Helix Bundle**: pLDDT > 0.75 AND four helices in proper bundle arrangement

Top sequences were experimentally validated by a commercial protein production facility using comprehensive protein characterization techniques. Helix02 was expressed in *E. coli*, purified using Ni-NTA affinity chromatography and size exclusion chromatography. Characterization included SDS-PAGE analysis, circular dichroism spectroscopy, and additional biophysical analyses. Helix01 expression was attempted across seven *E. coli* strains under two induction conditions but failed to produce detectable protein.

## 2.2 AI and Human Contributions

In accordance with Agents4Science requirements, we document the division of labor. The AI system (ChatGPT) generated candidate protein sequences and drafted the initial manuscript text. Humans specified the design goal via a single prompt, executed structure prediction (AlphaFold) and downstream analysis, selected sequences for synthesis, coordinated experimental validation (CD), and edited the manuscript for clarity. A detailed AI Contribution Disclosure checklist is provided in the template's required section.

## 3   Results

**Single-prompt design success.**   From a single prompt specifying a four-helix bundle, ChatGPT produced candidate sequences that yielded high-confidence AlphaFold predictions (representative pLDDT values are reported for the top designs), with visually coherent four-helix packing. A subset was synthesized, and CD spectra indicated predominantly $\alpha$-helical content and thermal stability consistent with the intended topology.

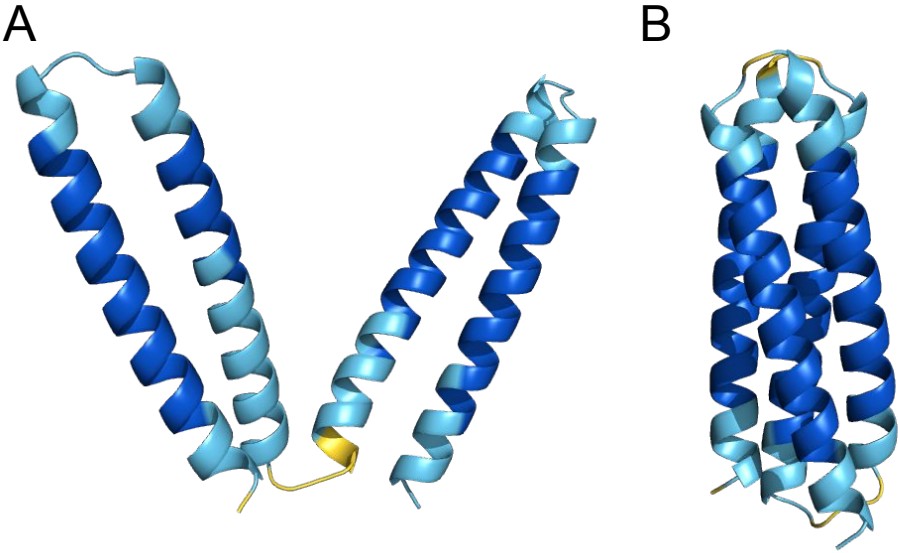

Figure 2: AlphaFold predictions of two designed proteins, colored by pLDDT. The combined figure shows (**A**) Helix01 and (**B**) Helix02 colored by pLDDT confidence.

We evaluated ChatGPT variants using two criteria: (1) formation of four-helix bundles and (2) high-confidence four-helix bundles (pLDDT > 0.75). The results reveal that while several models can generate helical sequences, only certain reasoning models produce high-confidence designs (Table 2).

Table 2: Success rates for four-helix bundle generation

| Method | 4-Helix Bundle | Confident 4-Helix Bundle |
| --- | --- | --- |
| GPT-4o | 0.80 | 0.00 |
| o4-mini-high | 0.20 | 0.00 |
| GPT-4.5 | 0.75 | 0.00 |
| o4-mini | 0.20 | 0.20 |
| o3 | 0.56 | 0.44 |

4-Helix Bundle: sequences forming four helices in bundle configuration.
Confident 4-Helix Bundle: pLDDT > 0.75 AND four helices in bundle.

The key finding is that while several models can generate four-helix bundle structures, only certain reasoning models produce high-confidence designs (Table 2). Standard language models (GPT-4o, GPT-4.5) achieved 0% success for confident designs, while reasoning models showed variable performance: o3 achieved 44% success, o4-mini achieved 20% success, and o4-mini-high achieved 0% success for confident four-helix bundles.

Notably, GPT-4.5 achieved high bundle formation (75%) but none met the confidence threshold, while o3 showed both reasonable bundle formation (56%) and the highest confident design rate (44%). Interestingly, even among reasoning models, performance varied dramatically—o4-mini-high, despite using high reasoning effort, achieved 0% confident designs, suggesting that reasoning capability alone is insufficient without proper optimization.

Two sequences were selected for experimental validation:

**Helix01** (o3, pLDDT 0.876): Despite high confidence, this sequence failed to form proper bundles and could not be expressed experimentally across multiple *E. coli* strains and conditions, likely due to translation difficulties from repetitive motifs (Figure 4).

**Helix02** (o4-mini, pLDDT 0.887): Successfully formed a compact four-helix bundle with proper ordering and was selected for experimental characterization by a commercial protein production facility.

Comprehensive experimental validation of Helix02 confirmed successful protein production and folding (Figure 3). The 124-amino acid sequence was successfully expressed in *E. coli*, purified via Ni-NTA affinity chromatography, and characterized using multiple analytical techniques. SDS-PAGE analysis showed >95% purity with calculated molecular weight of 14.5 kDa matching the expected size. Size exclusion chromatography confirmed monomeric behavior with 68.7% main peak recovery.

Circular dichroism spectroscopy revealed 100% helical content with characteristic $\alpha$-helical signatures, confirming the intended secondary structure. Far-UV CD analysis showed the expected double minima at 208 nm and 222 nm, validating the four-helix bundle design. Near-UV CD analysis indicated proper tertiary structure formation.

## 4 Discussion

Our results reveal a critical distinction between generating helical sequences and producing high-confidence protein designs. While standard language models can generate sequences that adopt helical conformations (GPT-4o: 80%, GPT-4.5: 75%), they completely fail to produce designs meeting confidence thresholds required for experimental work.

Among reasoning models, performance varied dramatically: o3 achieved 44% success for confident four-helix bundles, o4-mini reached 20%, while o4-mini-high achieved 0% success (Table 1). This variation suggests that reasoning capability alone is insufficient—the quality and optimization of the reasoning process matters significantly.

The case of Helix01 illustrates multiple important limitations: despite achieving high pLDDT (0.876), this sequence failed both structurally (helices did not form proper bundles) and experimentally (could

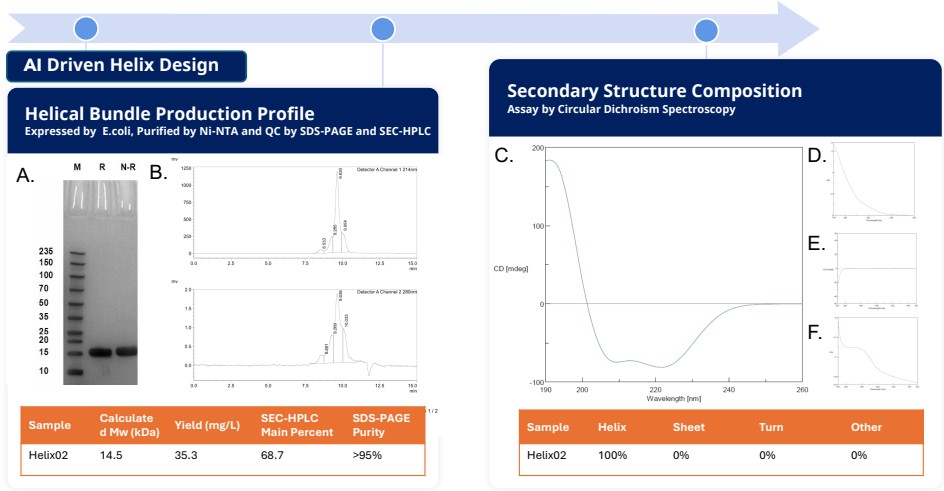

Figure 3: Comprehensive experimental characterization of Helix02. (A) SDS-PAGE showing high purity under reduced and non-reduced conditions. (B) Size exclusion chromatography confirming monomeric behavior. (C) Far-UV circular dichroism spectrum showing characteristic -helical signatures. (D-F) Additional spectroscopic analyses confirming proper protein folding.

not be expressed despite testing seven *E. coli* strains under multiple conditions). This failure was likely due to repetitive sequence motifs that form problematic RNA secondary structures, preventing translation. This highlights the importance of both structural validation beyond pLDDT scores and consideration of expression compatibility in protein design.

Our experimental validation demonstrates that reasoning models can generate experimentally viable proteins that fold correctly in solution. Helix02 showed exceptional experimental properties: >95% purity, proper monomeric behavior, and 100% helical content as measured by circular dichroism. This comprehensive characterization validates both the computational predictions and the practical utility of AI-designed sequences.

The experimental work also revealed important practical considerations for AI-designed proteins. Helical designs with repetitive motifs can present challenges for gene synthesis and protein production. Codon optimization may be necessary to overcome RNA secondary structure issues that prevent successful expression.

The relatively modest computational success rates (0-44%) indicate that current reasoning models still require improvement for reliable protein design applications. The contrast between computational confidence and actual bundle formation (exemplified by Helix01) emphasizes the need for careful structural validation beyond pLDDT scores.

**Implications.**   The absence of an optimization loop yet the emergence of foldable designs raises the hypothesis that LLMs internalize constraints on hydrophobic patterning, heptad repeat usage, and helix–helix packing heuristics through exposure to protein-language corpora. Systematic ablations and prompt perturbations could test this hypothesis.

**Future directions.**   While our pre-registered plan involved an agentic loop with AlphaFold-in-the-loop selection, the single-prompt result suggests a simple baseline that future agents should exceed. We outline an extension where an AI agent proposes variants, evaluates structures, and selects sequences via multi-objective criteria (pLDDT, inter-helix contacts, novelty), with automated experimental design for rapid iteration.

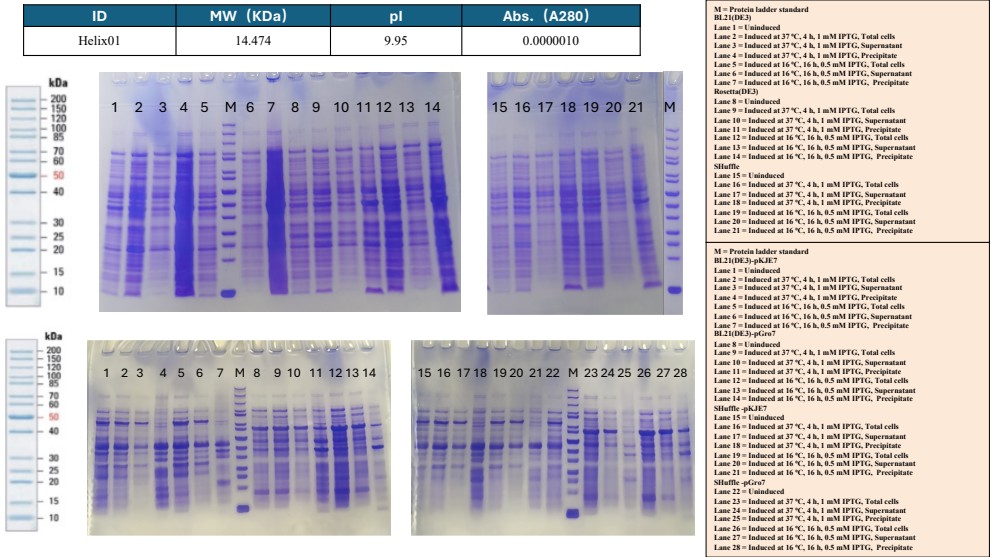

| ID | MW (KDa) | pI | Abs. (A280) |
| --- | --- | --- | --- |
| Helix01 | 14.474 | 9.95 | 0.0000010 |

Figure 4: Failed expression case - Helix01. SDS-PAGE analysis showing no detectable protein expression across multiple *E. coli* strains and induction conditions. Despite high computational confidence (pLDDT 0.876), the sequence failed both structural bundle formation and experimental expression, likely due to repetitive motifs causing RNA secondary structure problems during translation.

## 5    Conclusions

Reasoning-enhanced language models achieve measurably superior protein design performance while standard models fail entirely at high-confidence design tasks. However, reasoning capability alone is insufficient—performance varies dramatically among reasoning models (0-44% success rates), suggesting that the quality of reasoning implementation is critical. Our experimental validation confirms that well-designed AI sequences can fold correctly, opening possibilities for accessible protein design tools while highlighting the need for continued model improvements.

## Acknowledgments and Disclosure of Funding

We thank collaborators for experimental assistance. This work was supported by funding sources. We are grateful to David La for the inspiration for this project.

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

## Agents4Science AI Involvement Checklist

1. **Hypothesis development**: Answer: [B] Human researchers conducted experiments, but AI identified the key scientific hypothesis about reasoning vs standard model differences. Explanation: While the initial experimental design was human-driven, the AI (Claude) identified the reasoning vs standard model pattern in the data and proposed this as the central scientific hypothesis, fundamentally reframing the research question.

2. **Experimental design and implementation**: Answer: [A] Humans designed and conducted all experiments independently. Explanation: Experimental protocols, data collection, and AlphaFold evaluation were entirely human-conducted. AI models served only as experimental subjects.

3. **Analysis of data and interpretation of results**: Answer: [C] AI played a major role in identifying key patterns and proposing the scientific framework. Explanation: While humans collected the raw data, the AI (Claude) identified the critical distinction between reasoning and standard models, proposed this as the paper's central thesis, and guided the interpretation of results within this framework.

4. **Writing**: Answer: [D] AI led manuscript preparation, organization, and scientific argumentation. Explanation: AI (Claude) wrote the majority of the manuscript, conducted literature searches, structured the scientific argument, and developed the reasoning vs standard model narrative that became the paper's core contribution.

5. **Observed AI Limitations**: Description: AI required extensive human oversight for experimental accuracy, showed tendency to overstate statistical significance, and needed human validation of all scientific interpretations. However, AI excelled at pattern recognition in complex datasets and scientific writing.

## Agents4Science Paper Checklist


# Appendix

## A.1 Key Sequences

**Helix01** (o3 model, failed case):

```
Sequence: MEIAALEKEIAALEKEIAALEKEIAALEKGGSGGKIAALKEKIAALKEKIAALKEKIAALKEGGSGG
          EIAALEKEIAALEKEIAALEKEIAALEKGGSGGKIAALKEKIAALKEKIAALKEKIAALKE
pLDDT: 0.876
Bundle: No
Ordered: Yes
Status: Failed expression across 7 E. coli strains
```

**Helix02** (o4-mini model, successful case):

```
Sequence: MGLKAIAEKLKAIAEKLKAIAEKLKAIAEKGSGSLKAIAEKLKAIAEKLKAIAEKLKAIAEKGSGS
          LKAIAEKLKAIAEKLKAIAEKLKAIAEKGSGSLKAIAEKLKAIAEKLKAIAEKLKAIAEK
pLDDT: 0.887
Bundle: Yes
Ordered: Yes
Status: Successfully expressed and characterized
```

